# Effects of Injection Molding Parameters on Properties of Insert-Injection Molded Polypropylene Single-Polymer Composites

**DOI:** 10.3390/polym14010023

**Published:** 2021-12-22

**Authors:** Jian Wang, Qianchao Mao, Nannan Jiang, Jinnan Chen

**Affiliations:** 1State Key Laboratory of Organic-Inorganic Composites, Beijing University of Chemical Technology, Beijing 100029, China; 2College of Mechanical and Electrical Engineering, Beijing University of Chemical Technology, Beijing 100029, China; 3School of Chemistry and Chemical Engineering, Beijing Institute of Technology, Beijing 100081, China; maoqianchao@126.com (Q.M.); 2120141212@bit.edu.cn (N.J.); jnchen@bit.edu.cn (J.C.)

**Keywords:** injection molding, processing parameter, single-polymer composite, polypropylene, mechanical properties

## Abstract

The reinforcement and matrix of a polymer material can be composited into a single polymer composite (SPC), which is light weight, high strength, and has easy recyclability. The insert injection molding process can be used to realize the multiple production of SPC products with a short cycle time and wide processing temperature window. However, injection molding is a very complicated process; the influence of several important parameters should be determined to help in the future tailoring of SPCs to specific applications. The effects of varying barrel temperature, injection pressure, injection speed, and holding time on the properties of the insert-injection molded polypropylene (PP) SPC parts were investigated. It was found that the sample weight and tensile properties of the PP SPCs varied in different rules with the variations of these four parameters. The barrel temperature has a significant effect, followed by the holding time and injection pressure. Suitable parameter values should be determined for enhanced mechanical properties. Based on the tensile strength, a barrel temperature of 260 °C, an injection pressure of 127.6 MPa, an injection speed of 0.18 m/s, and a holding time of 60 s were determined as the optimum processing conditions. The best tensile strength and peel strength were up to 120 MPa and 19.44 N/cm, respectively.

## 1. Introduction

Single-polymer composite (SPC) is a composite material whose reinforcement and matrix are from the same type of polymer. SPC possesses the advantages of low density, good interfacial compatibility, high strength, and easy recyclability [1,2,3]. Currently, SPC is becoming of high interest in the field of composites as well as commercial and industrial applications [4]. The preparation methods of SPC products are mainly based on compression molding [1,2,3,4], such as the hot compaction of polymer fibers/tapes/fabrics, film stacking, co-extrusion combined with compression. The compression molding method for SPC production has many disadvantages, such as a narrow temperature window, the difficult controlling of processing temperature, a long molding cycle, and low efficient production. In order to overcome these disadvantages, injection molding is a potential method to be used in the preparation of SPCs. Injection molding is an effective method for the rapid batch molding of plastic products. It plays an important role in the molding of the products of short/long glass/carbon/nature fiber reinforced polymer composites [5]. However, the traditional injection molding method cannot be used in the production of SPC parts. The reinforcement of the same polymer as the matrix material would be molten during the injection molding processing, and thus the reinforcement effect would disappear. The establishment of a processing temperature window which is the melting temperature difference between the reinforcement and the matrix, is the key to realizing the production of SPC parts. Kmetty et al. [6,7] applied highly oriented homo polypropylene (PP) as the reinforcement and a PP-based thermoplastic elastomer as the matrix, and firstly prepared SPC pellets (5 mm × 5 mm) by film lamination method and cutting, and finally added the SPC pellets into the injection molding machine to produce SPC parts. The results showed that the processing temperature window was expanded to 90 °C due to the addition of the PP elastomer. However, the elastomer and the cut fibers decreased the improvement of the mechanical strength of the PP SPC parts. Andrzejewski et al. [8] combined co-extrusion and injection molding methods to prepare PP SPCs. The processing temperature window of 33 °C was realized. However, the method is not the best solution in terms of high mechanical properties due to the short fibers and their anisotropic distribution. Khondker et al. [9] applied injection-compression molding for the production of PP SPC parts. The weft-knitting technique was conducted to prepare plain knitted textile fabrics as the reinforcement. Before the injection-compression molding, the fabrics should be fixed in the mold. The short cycle time could be achieved by this method in comparison with compression molding. Unfortunately, they mainly discussed the processing temperature and the effects of homo-PP and block-PP and did not discuss the effects of other processing parameters. Moreover, the tensile strength of the prepared PP SPCs was not improved. A similar approach for preparing SPCs by insert injection molding was proposed using PP and polyethylene (PE) as models [10,11,12]. The method has a relatively high production efficiency with a molding cycle on the order of seconds. The processing temperature window for PP SPCs and PE SPCs achieved 80 and 40 °C, respectively. A sandwiched structure with a middle fabric for PP and PE SPCs was also realized by the insert injection molding method [13,14]. This method can realize a better appearance because the fabric was set in the middle of the parts. However, the fiber volume fraction was low, which influenced the improvement of the mechanical properties of the SPC parts. Andrzejewski et al. [15] applied the same method in the preparation of poly(ethylene terephtahalte) (PET) SPCs. The presented results confirm that the proposed concept of using the overmolding technology for the preparation of SPCs has the potential for industrial implementation. However, the improvement in the tensile strength of the SPCs was also limited. Jerpdal et al. [16] has also studied the influence of temperature on the properties of the insert injection molded PET SPCs. They pointed out that a thorough knowledge of how temperature influences the SPCs is required. A melt sequential injection molding process [17] was developed to form a composite structure of SPCs, but the improvement of tensile strength was constrained by the limited met injection number. Similarly, co-injection molding was used in the preparation of SPCs [18,19]. However, a specially designed mold was needed, and the melt sequential injection molding process increased the complication. Thus, the method is not suitable for common plastic products. Furthermore, a very limited increase in tensile strength can be achieved. 

The process parameters for hot compacted SPCs, compression-moulded SPCs, and co-extruded SPCs have already been investigated [20,21,22,23]. In comparison, injection molding is more complicated. The molding time in minutes and seconds involves the joint action of multiple process parameters. There are several parameters that should be controlled in injection molding, including the barrel temperature, injection speed, injection time/volume, injection speed, injection/holding pressure, packing/holding time, and cooling time. Not all of these parameters have a significant influence on the properties of final SPC products. A previous study confirmed that barrel temperature was the most important parameter [10,11,12,13,14,16], but the influence of other parameters can vary across the whole injection molding cycle depending on the material, equipment, and mold cavity. It is important to understand the influences of these process parameters for injection molding of SPCs. The aim of this work is to analyze the influence of various process parameters on the properties of the insert injection molded PP SPCs. Four important injection molding parameters, including barrel temperature, injection pressure, injection speed, and holding time, were selected as the variable processing parameters, and their effects on the weight and mechanical properties of the insert injection molded SPC samples were investigated. The results can be applied to guide the multiple production processes of SPC products at the industrial level. 

## 2. Materials and Methods

### 2.1. Materials

PP granules (Marlex HGZ-1200, Phillips Sumika Polypropylene Company, The Woodlands, TX, USA) with a density of 0.907 g/cm^3^ and a melt flow rate of 115 g/10 min at 230 °C were used as the matrix. A commercial plain-woven PP fabric supplied by Innegrity LLC (Simpsonville, SC, USA) was selected as the reinforcement. The plain-woven fabric had an areal density of 170 g/m^2^ (8 threads/cm in warp and weft directions) and a thickness of approximately 0.44 mm. Each bundle consisted of 225 individual fibers. The single fiber had a diameter of 48 μm. A tensile strength of 560 MPa and a tensile modulus of 6.6 GPa of the single fiber was determined by a universal tensile test machine (5166, Instron Corp., Norwood, MA, USA). The melting points of the PP matrix and fiber tested by a differential scanning calorimeter (Q200, TA Instruments, New Castle, NY, USA) were 166 °C and 177 °C, respectively. Table 1 present the main information of the materials used in the preparation of PP SPC samples. 

### 2.2. Rheology Measurement for the PP Matrix

A capillary rheometer (LCR7000, Dynisco Co., Franklin, MA, USA) was employed to measure the rheological properties of the matrix, in particular, the viscosity and the shear rate in reference to the process temperature. PP pellets were fed in the barrel of the rheometer and maintained at the set temperature for 15 min. The orifice diameter of the capillary was 0.5 mm. The apparent viscosity of the matrix melt under different shear rates was collected, four constant temperatures (200, 220, 240, and 260 °C) were used, respectively. 

The shear rate, which corresponded to the matrix melt through the gate of the mold cavity, was determined according to the injection flow rate and the gate dimensions of the mold cavity. The apparent shear rate through a rectangular gate could be calculated according to the formula
(1)$$\dot{\gamma }=\frac{6Q}{W_{gate}\cdot H_{gate}^{2}}$$
(2)$$Q=\frac{V_{cavity}}{t_{filling}\cdot n_{gate}}$$
where *W_gate_* is the gate width (1.4 mm), *H_gate_* is gate thickness (0.52 mm), *Q* is the polymer flow rate through the gate, *V_cavity_* is the volume of the mold cavity (723.4 mm^3^), *t_filling_* is the filling time (1 s), *n_gate_* is the gate number which is one in this case. The calculated shear rate was 11,473.48 s^−1^ and can be used to analyze the exact viscosity at the gate at different temperatures. 

### 2.3. Preparation

The injection molding machine (SE-18D, Sumitomo Co.) was used in the insert injection molding. Figure 1 show the schematics of the mold cavity and the insert injection molding structure of the SPC sample. Figure 2 show the mold plate, PP fabric, and the tensile specimen of PP SPC after the tensile test. The mold used in the experiments contained a rectangular cavity of dimensions 63.5 mm × 9.5 mm × 1.2 mm and was maintained at a temperature of 20–30 °C. During the process, the PP woven fabric was firstly cut into layers of 63.5 mm × 9.5 mm, as shown in Figure 2. Two weave layers were used as the reinforcement, and each side of the mold cavity walls was attached by one layer. A small number of double-sided tapes assisted the fixing of the inserted fabric. After the previous preparation of inserted fabrics, injection molding can be conducted. Under different injection molding processes, the melted PP matrix was pushed from the barrel to fill the mold cavity and permeate the reinforcement. Finally, the SPC part was ejected together with the two layers of fabrics from the mold. 

Four important injection molding parameters, including barrel temperature, injection speed, injection pressure, and holding time, were selected as the variable processing parameters. Considering the effects of these injection molding parameters are complex and nonlinear, the single-factor experimental design was applied, as listed in Table 2. Barrel temperature ranged from 200 to 280 °C (Experimental No. 1–5). The injection pressure limitation of the injection molding machine was 30 kpsi. Since the cavity cannot be filled completely below 5 kpsi, 5~30 kpsi were chosen as the variation range of the injection pressure. The setting injection pressure was 5, 10, 15, 20, 25, and 30 kpsi. The real injection pressures measured by the machine sensor were 6.5, 8.5, 10.5, 13.5, 16.5, and 18.5 kpsi, corresponding to 44.8, 58.6, 72.4, 93.1, 113.8, and 127.6 MPa, respectively (Experimental No. 6–11). The injection speed was changed from 1 to 11 in/s (0.03 to 0.28 m/s) (Experimental No. 12–17). The holding time was changed from 1 to 120 s (Experimental No. 18–25). The holding pressure was at 80% of injection pressure and changed with the injection pressure. The other processing parameters were kept constant. The filling time and cooling time were set to 1 s and 10 s, respectively. The optimal parameter values (Experimental No. 26) were determined by the optimal tensile strength of the PP SPC samples in each group of the single-factor experiments. 

### 2.4. Weight Measurement

The weight of each sample was measured by an electronic balance (AUW120D, Shimadzu) with an accuracy of 0.001 g.

### 2.5. Mechanical Tests

The tensile tests were carried out on a universal testing machine (Instron 5166). The tensile sample was cut from the rectangular SPC sample by a dumbbell-shaped cutter. The dumbbell shape follows the testing specimen under the guidelines of DIN-53504. The crosshead speed was 5 mm/min with a gauge length of 5 mm. The load cell was electronically scaled to 30 kN. Each test was performed at room conditions (around 25 °C), and at least six samples were tested for each group.

### 2.6. Peel Tests

To evaluate the interfacial adhesion, the injection volume was decreased, and other injection molding conditions remained unchanged, then the uncompleted filling could be obtained. As the melt had not fully filled the cavity, the flow end was approximately 1 cm away from the end of the cavity. The sample could be used for the T-peel test. Both ends of the opened sample were clamped, respectively, by the two collets of the universal testing machine (Instron 5166). One layer of fabric in the sample was used as one end of the crack, and the other layer of fabric and PP matrix were used as the other end of the crack. The sample was pulled apart at a crosshead speed of 20 mm/min. The relationship data between tension and displacement was recorded. The test sample was 62.5 mm long and 9.3 mm wide. The average peel force was calculated for each peak of the weighted average tensile force and displacement curve.

### 2.7. Digital Microscopy

A digital microscope (VHX-2000, KEYENCE Deutschland GmbH, Neu-Isenburg, Germany) was used to observe the tensile sample of SPC prepared under the optimal processing conditions. The sample was initially cut at the cross-section and then photographed. The fracture section of the sample after the tensile test was also observed. 

## 3. Results and Discussion

### 3.1. Effect of Barrel Temperature

Injection temperature is a crucial parameter for the insert injection molded PP SPCs. A lower injection temperature may cause bad flowability and a short shot. A higher temperature would cause the fiber to melt. In order to know the exact effect of temperature on the flowability of the PP matrix, rheology measurement was conducted. Figure 3 show the apparent viscosity of the PP matrix changing with the shear rate at different temperatures. At a certain shear rate, the viscosity of the PP matrix decreases with the increase of temperature. For PP melt, the relationship between viscosity and temperature approximately conforms to the Arrhenius equation, which is:(3)$$\mu =Ae^{E/RT}$$
where *A* is the constant representing the characteristics and relative molecular weight of the polymer under given shear stress, *E* is the flow activation energy characterizing the temperature dependence of the melt shear viscosity, *R* is the gas constant, and *T* is the absolute temperature. Since the flow activation energy *E* of PP is in the range of 37.5~62.7 kJ/mol, the flow activation energy is relatively small. In the region of the mold gate, the injection filling flow of PP melt is mainly molecular flow. As the PP resin selected in this experiment is ultra-high fluidity PP, it has a high melt flow index, low viscosity, and good flow performance. The shear rate of 11,473.48 s^−1^ corresponds to the viscosity of 19, 16, 14, and 13 Pa·s at temperatures of 200, 220, 240, and 260 °C, respectively. Higher temperatures cause lower viscosity which benefits the injection filling and the impregnation of matrix melt into the gaps of the fabrics (as shown in Figure 1). 

Figure 4 show the weight, tensile strength, and tensile modulus of the PP SPC samples prepared at different barrel temperatures. The other parameters were kept constant while the barrel temperature varied from 200 to 280 °C when the PP SPC samples with double-layered fabrics were manufactured in Experimental No. 1–5 (Table 2). In Figure 4a, the weight of the PP SPC sample increased continuously at higher barrel temperatures. During the infiltration procedure of the matrix in the fabric, there are two kinds of flow scales. Macroscopic flow fills the space between the fiber bundles. The other, microscopic flow, enables the matrix to flows through the space among single fibers. The microscopic flow belongs to a smaller flow size which is directly slow and difficult to flow for impregnation. Figure 3 show that higher temperature would reduce the viscosity of the matrix, then the permeability would be improved. Therefore, the improved melt fluidity promoted the matrix to penetrate into the space in the bundles and fibers. Before the matrix cooled down and lost its fluidity, the mass of the matrix penetrating into the fabric increased, thus higher barrel temperature led to higher sample weight. It can also be seen in Figure 4a that there are two abrupt increases of sample weight before 210 °C and after 260 °C. The sample weight increased by 7.6% from 200 to 210 °C and 12.8% from 260 to 270 °C. This increasing trend is in accordance with the decreasing trend of the viscosity as a function of temperature. The first abrupt increase indicates bad infiltration due to much lower temperature, while the second abrupt increase indicates fiber melting. When the barrel temperature increased above 270 °C, the sample weight presented a substantial promotion of 12.8% compared with that at 200 °C. The reason was that a higher barrel temperature resulted in more melted fibers, which could form a PP matrix with higher density compared with PP fabric under the compaction of the injection pressure. During injection molding, the cooling rate was extremely fast in the mold cavity, the time of matrix infiltrating fabrics was very short, and the decreased viscosity at the temperature lower than 260 °C did not significantly improve the wettability.

Figure 4b show that the tensile strength and modulus of PP SPCs increased then decreased with the increasing barrel temperature. The maximum values of the tensile strength and modulus of PP SPCs were 103 MPa and 960 MPa, respectively. Higher barrel temperature resulted in lower viscosity, which is beneficial for infiltration, and thus the interfacial adhesion between matrix and fabric was improved. At the same time, the tensile strength and modulus were enhanced. However, when the barrel temperature reached 280 °C, the tensile strength and modulus went down rapidly, indicating that the excessively-high barrel temperature made the fibers partially melt. It is noted that the corresponding barrel temperature for tensile strength was 260 °C which is in accordance with the temperature for the abrupt increase of sample weight. The transfer temperature for the tensile modulus was 240 °C, which is lower than that for tensile strength. This was related to the molecular relaxation, and the plain weaved structure of the fabric. The flexion structure of the fiber bundles increased the strain under tensile load, although the stress was still high when the barrel temperature was 240 °C. At 280 °C, the tensile strength of PP SPC was still higher than that of pure the PP matrix (around 30 MPa), illustrating that some fibers can still maintain their structural integrity and mechanical properties and play the role of enhancement. As the inserted fabrics were fixed in the mold cavity, the reinforced form of the fabric did not flow with the matrix melt even at a high barrel temperature. 

### 3.2. Effect of Injection Pressure

Injection pressure refers to the pressure exerted by the top of the screw to push the melt to fill the mold cavity and compact the melt. It is one of the key parameters affecting the product properties. When the injection pressure was too high, it would result in a few undesirable phenomena such as inflation and overflow, and larger pressure fluctuations would generate bubbles and silver streaks inside products. When the injection pressure was relatively low, the melt could not fill the cavity completely. 

Figure 5 show the sample weight, tensile strength, and modulus of PP SPCs prepared at different injection pressures. The other processing parameters were set as constants in Experimental No. 6–11 (Table 2). As shown in Figure 5a, with the injection pressure increasing, the sample weight increased at first because the maximum pressure in the cavity became larger at higher injection pressure, and more PP matrix penetrated into the fiber interspaces. When much higher injection pressure was used, there was no increase in sample weight and it even decreased less. This indicates that excessively high pressure might result in closer fiber arrangement, impeding further penetration of the matrix. 

In Figure 5b, the tensile strength and modulus increased continuously with the increasing injection pressure. The relatively high injection pressure was beneficial to matrix infiltration inside the fabric. A larger interfacial bonding area to improve the tensile properties could be formed. When the injection pressure was at 127.6 MPa, the tensile strength and modulus reached the maximum values which were 94 MPa and 1012 Mpa, respectively. Since the density of the PP resin was higher than that of the PP fibers, the interspace that existed in the fibers was replaced by the PP resin under the pressure effect; thus, there was a great improvement in the quality of the composites.

### 3.3. Effect of Injection Speed

When the melt was injected into the cavity through the nozzle and gating system during the high-speed filling process, there would be abundant frictional heat generated in this process to improve the melt temperature. When the mold cavity was filled at a high injection speed, the welding of the insert rear was not satisfied which caused the product strength to reduce. When the cavity was filled at a low injection speed, the filling time was longer. The melt injected into the cavity initiallycooled down rapidly and presented greater viscosity, which caused higher pressure required for the subsequent filling process. That was why the uneven filling results occurred. Therefore, choosing the appropriate injection speed was one of the most important parameters to control the properties of PP SPC products.

In Figure 3, the apparent viscosity decreased rapidly with the increasing shear rate. When the injection speed was elevated, the filling time would shorten, and the shear stress endured by the melt would increase, indicating that increasing injection speed could decrease the melt viscosity. 

In the experiments (Experimental No. 12–17 in Table 2), a barrel temperature of 240 °C and injection pressure of 13 kpsi (89.6 MPa) were kept unchanged and the injection speed was varied: 1, 3, 5, 7, 9, 11 in/s, i.e., 0.025, 0.076, 0.13, 0.18, 0.23, 0.28 m/s. Figure 6 show the sample weight, tensile strength, and modulus of PP SPCs versus the injection speed. As the injection speed increased, the filling time was shortened, the melt flow rate increased, and the melt endured higher shear stress. The melt temperature increased under the effects of shear stress and friction heat, which was beneficial for the matrix melt to penetrate into fabrics and thus improved the sample weight and tensile strength. Moreover, the melt could be injected into the cavity to make the melt pressure reach the maximum value in a short time at a high injection speed. During the procedure, the melt possessed lower viscosity and excellent permeability. As the barrel temperature of 240 °C was enough and the increase of temperature by shear rate was limited, the tensile strength almost kept no change when the injection speed was lower than 0.18 m/s. The tensile strength reached the top value at the injection speed of 0.18 m/s. When the injection speed was further increased, the melt temperature increased much more due to the extra frictional heat generated when the melt went through the nozzle and runners; many more fibers were melted by the excessively high matrix temperature and lost their physical form and mechanical properties. The continuous decrease of tensile modulus was still mainly related to the fabric structure. The stretching process resulted in a larger strain which led to modulus reduction due to the plain-weaved structure of the fabric. In conclusion, considering the range of injection speed for these experiments, the mechanical properties of PP SPC achieved the optimal tensile strength at the injection speed of 0.18 m/s. Thus, 0.18 m/s was the most appropriate injection speed in this work.

### 3.4. Effect of Holding Time

The packing/holding stage refers to the period from the matrix filling the cavity to the screw beginning to recede. During this stage, the melt would be cooled down to a contractile. However, the screw was moved forward to promote the matrix to be injected into the cavity and supplemented the space generated through cooling shrinkage to maintain the melt pressure inside the cavity constant. The packing stage has great significance in improving the sample density, reducing shrinkage, and overcoming defects on surfaces. 

In Experimental No. 18–25 (Table 2), the processing parameters, including barrel temperature 240 °C, injection pressure 15 kpsi (103.4 MPa), and injection speed 0.13 m/s, were kept as constants, the holding time was then changed: 1, 5, 10, 15, 20, 25, 60, 120 s. Figure 7 show sample weight, tensile strength, and modulus of the PP SPCs changing with the holding time. The sample weight increased first and then tended to remain the same. The reason was that the material inside the cavity might shrink due to the cooling effect after the filling stage. Therefore, the longer the holding time, the more melt injected into the cavity, then the PP SPC sample possessed higher weight. When the holding time continued to extend to more than 60 s, the melt at the gate and inside the cavity had been cooled down completely. The solid material prevented the matrix from flowing into the cavity; thus the part weight remained changeless. 

Both of the tensile strength and modulus also improved at first then remained constant. After the filling stage, more of the matrix was injected into the cavity by moving the screw forward, and the melt pressure stayed the same, which could facilitate the matrix into the fibers and combine with them forming a superior interface, which could be beneficial to elevating the mechanical properties. Further extension of holding time might make the melt in the cavity cooled and solidified, then the matrix was hindered from going into the cavity, and the pressure stayed the same. Therefore, if there was no influence on the infiltration and combination with fibers along the extended holding time, then the mechanical performance of PP SPC would remain changeless. Considering high productive efficiency, 60 s was determined as the optimum holding time. 

### 3.5. Optimum Processing Condition

According to the above single-factor experimental analysis, the optimum processing condition for good mechanical properties of the insert-injection molded PP SPC could be determined as follows: barrel temperature of 260 °C, injection pressure of 127.6 MPa, injection speed of 0.18 m/s, and holding time of 60 s. By using this processing condition, the PP SPC sample was produced. For comparison, pure PP was also produced with the same processing conditions. Typical tensile stress–strain behaviors of the different tensile samples are shown in Figure 8. The insert injection molded PP SPC sample exhibited significant improvements in tensile strength compared to the pure PP sample. The optimum tensile strength was up to 120 MPa, which is 3.9 times that of the pure PP. In the case of pure PP, it exhibited high extensibility indicating elastic deformation due to its high ductility. In comparisin, the PP SPC presented plastic deformation, and embrittlement occurred. The obtained tensile strength of 120 MPa was also the highest value in all experiments, which proves the feasibility of the single-factor experimental design. 

The comparison of optimal tensile strength of PP SPC samples produced by different injection molding methods is listed in Table 3. The mechanical strength of SPCs was influenced by basic material strength, composite structure, and processing conditions. The tensile strength of PP was around 30 MPa [24]. Thus, the tensile strength under 40 MPa was not obvious. The composite structure includes the type of reinforcement, the volume fraction of fibers, the diameter and length of fibers, and fiber orientation or arrangement. Continuous fiber was usually used, and plain fabric was used to confirm the multidirectional bearing capacity. The fiber volume fraction and arrangement usually affect the processing. In this work, we firstly used plain fabric to confirm a good reinforcement type and then two layers of fabric to confirm a high fiber volume fraction. Finally, the complex processing conditions were optimized by the single-factor experiments. The comparison results showed a significant improvement via the insert injection molding with optimized processing conditions.

Figure 9 show the peel load curve of the PP SPC. The sawtooth trace was related to the fabric structure. Moreover, the peel load increased with the extension, which was related to the temperature distribution along the filling direction. During the insert injection molding, the PP matrix filled the cavity from the gate and infiltrated the fabric from one side of the fabric layer. Then the matrix cooled down rapidly due to the heat conduction around the cavity wall. The melt far from the gate had a lower temperature and higher viscosity, which made it more difficult for the matrix to penetrate into the interstices of the fabrics. The section far from the gate did not provide good interfacial bonding, and that is why the peel load went up gradually from the cavity end to the gate. In addition, the peel sample was prepared by reducing the injection volume, which led to incomplete filling and lower pressure inside the cavity. Accordingly, the matrix permeability was weakened, and so was the interfacial adhesion. The average value of the peel strength of the PP SPC could be up to 19.44 N/cm, indicating good interfacial bonding. 

The morphology of the sample prepared at the optimal processing conditions is shown in Figure 10. The SPC sample was cut at the cross-section, as shown in Figure 10a. Knife marks and a small number of pollutants can be seen on the cross-section. The results show good consolidation of the SPC sample. The fabric and matrix can still be distinguished. The dash lines indicate the interfaces between the matrix and the fabric. The overall adhesion bonding between the fabric and matrix is good. Figure 10b show the fracture section of the sample after the tensile test. Part of the matrix can be seen between the fibers, indicating that the matrix can impregnate into the gaps between fibers during the injection molding process. The interfacial adhesion between fibers can be guaranteed. Under the action of tensile force, there is an obvious boundary separation between the fiber and matrix. Microfibers can also be observed, indicating good interfacial adhesion.

## 4. Conclusions

Four injection molding parameters were especially exploited for the production of insert injection-molded PP SPC parts: barrel temperature, injection pressure, injection speed, and holding time. The effects of the four parameters on the weight and mechanical properties of the insert injection-molded PP SPCs products were investigated, respectively. Barrel temperature was the most important parameter. Higher barrel temperature led to excellent adhesiveness between the fabrics and the matrix and created a better bonding effect on the interfaces between matrix and fibers. However, a much higher barrel temperature will lead to fiber melting and thus reduce the mechanical properties of the SPC. Holding time and injection pressure were the second most important parameters. A longer holding time and higher injection pressure can make the matrix fully fill the cavity and impregnate the fabric, but when the gate freezes, the additional holding time and injection pressure will lose their effects. Injection speed had a great influence on the filling process of the insert injection molding, but it had no significant effect on the properties of SPC samples. Considering beneficial bonding and enhancement from the fabric, suitable injection molding parameters should be chosen. According to the tensile strength of PP SPC products, the four parameters were chosen for the optimum processing conditions: barrel temperature of 260 °C, injection pressure of 127.6 MPa, injection speed of 0.18 m/s, and holding time of 60 s. Experimental results showed that the tensile strength of prepared PP SPC at the optimum processing parameters was 120 MPa, 3.9 times of the unreinforced PP. This shows a significant improvement in the tensile strength of PP SPCs in comparison with other injection molding techniques for PP SPCs. The average peel force of the PP SPC was 19.44 N/cm, indicating superior interfacial bonding. However, the optimum processing condition in this work is not suitable for other geometry, polymer materials, and injection molding machines. The results in this work can be used to guide the mass injection molding production of other SPC products of different materials with different geometry, especially for the adjustment of injection molding parameters. Future work would focus on numerical simulation to verify the influence law of multi-parameters on the insert injection molding of SPC parts. Furthermore, a method to improve the surface quality of SPC parts would also be developed. 

## Figures and Tables

**Figure 1 polymers-14-00023-f001:**
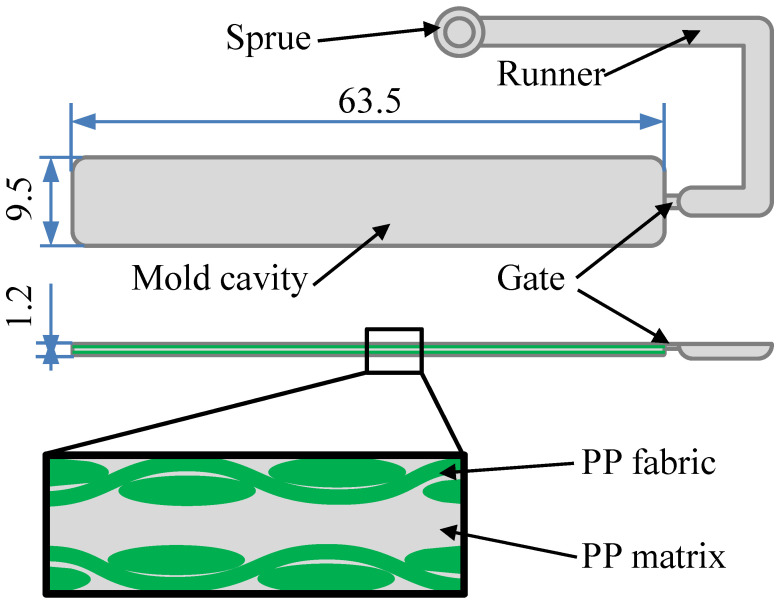
Schematics of the mold cavity and the insert injection molding structure of the PP SPC sample.

**Figure 2 polymers-14-00023-f002:**
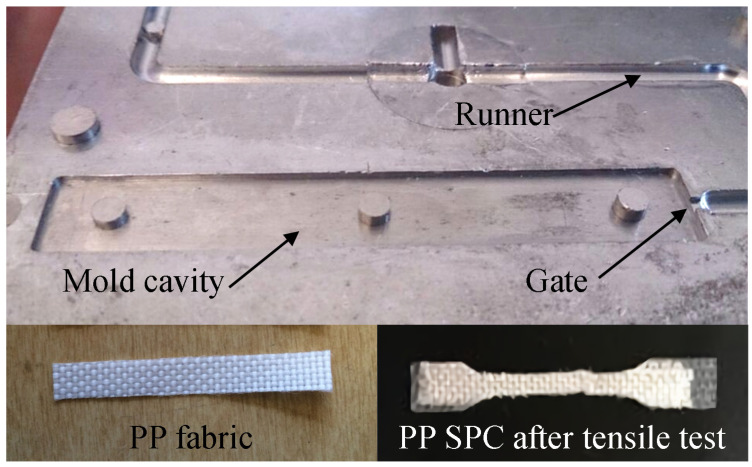
Pictures of the mold plate, PP fabric, and the PP SPC after tensile test.

**Figure 3 polymers-14-00023-f003:**
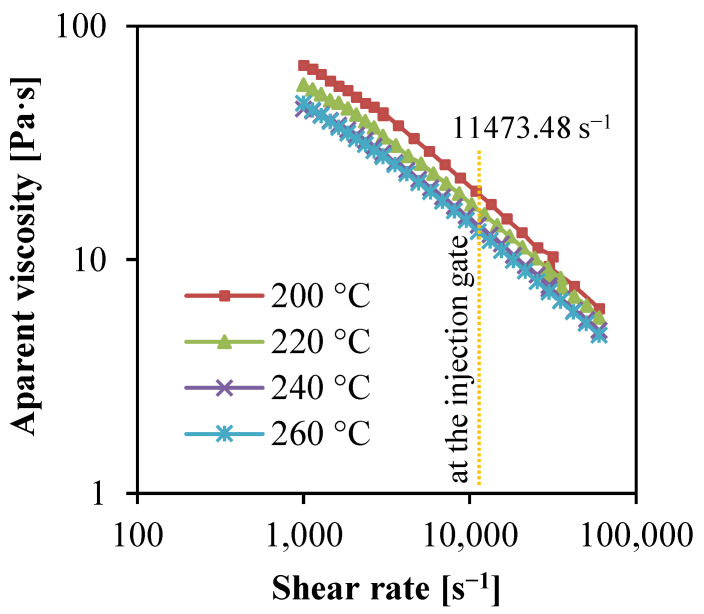
Apparent viscosity of PP matrix changing with the shear rate at different temperatures.

**Figure 4 polymers-14-00023-f004:**
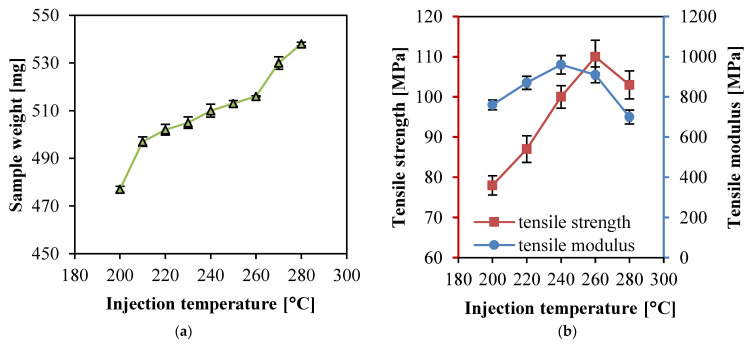
Properties of the PP SPC samples prepared at different barrel temperatures: (**a**) weight, and (**b**) tensile and tensile modulus.

**Figure 5 polymers-14-00023-f005:**
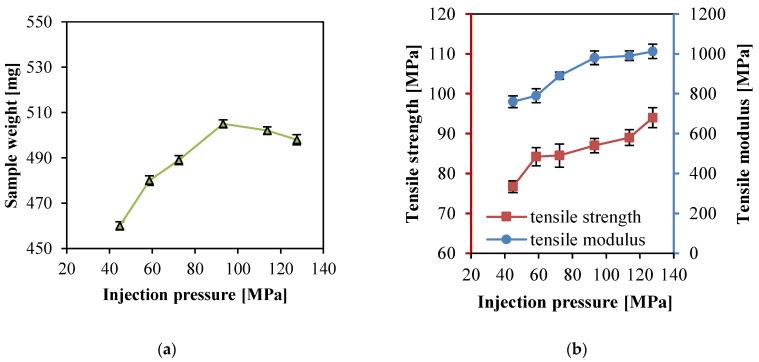
Properties of the PP SPC samples prepared at different injection pressure: (**a**) weight, and (**b**) tensile and tensile modulus.

**Figure 6 polymers-14-00023-f006:**
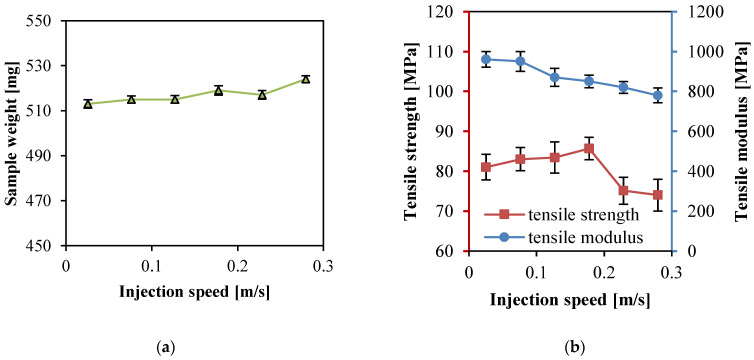
Properties of the PP SPC samples prepared at different injection speeds: (**a**) weight, and (**b**) tensile and tensile modulus.

**Figure 7 polymers-14-00023-f007:**
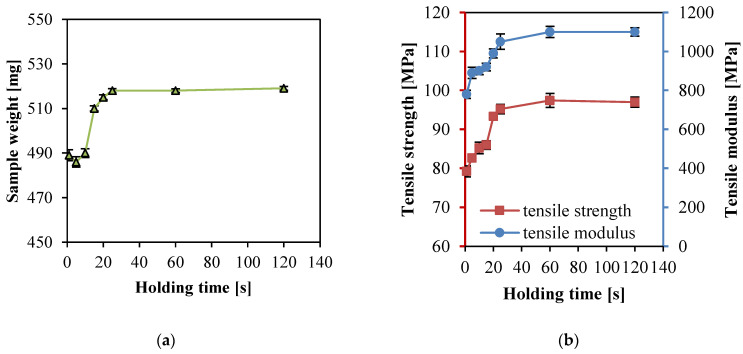
Properties of the PP SPC samples prepared with different holding times: (**a**) weight, and (**b**) tensile and tensile modulus.

**Figure 8 polymers-14-00023-f008:**
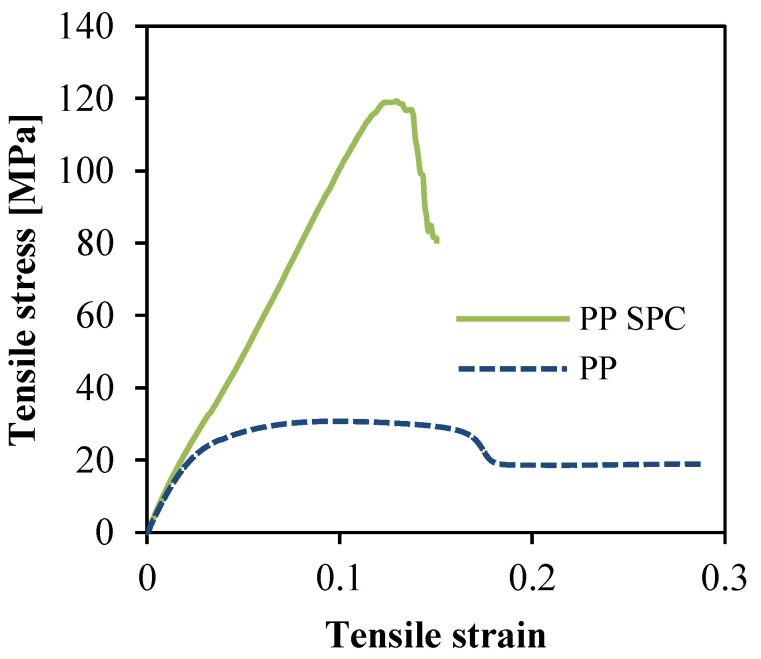
Tensile stress–strain behaviors of the different tensile samples of PP SPC and pure PP.

**Figure 9 polymers-14-00023-f009:**
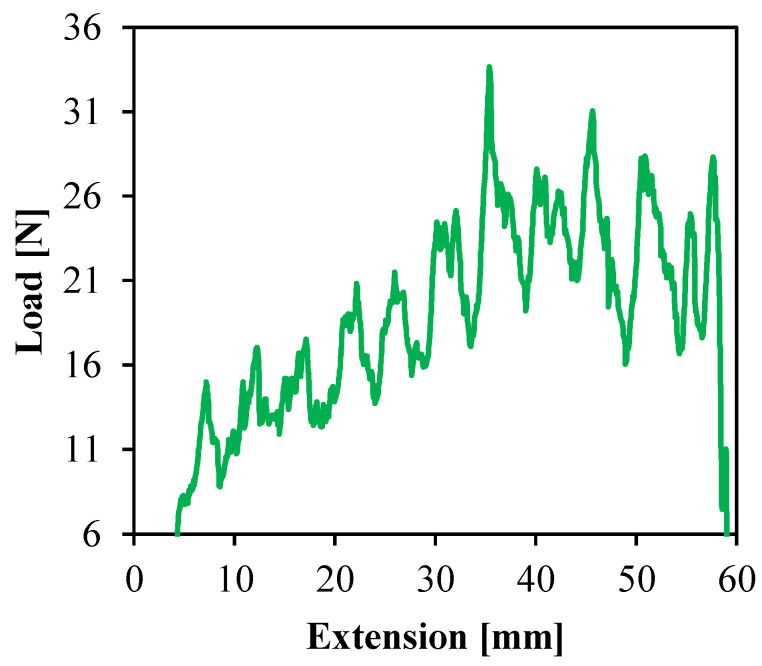
Peel load curve of the PP SPC as a function of extension.

**Figure 10 polymers-14-00023-f010:**
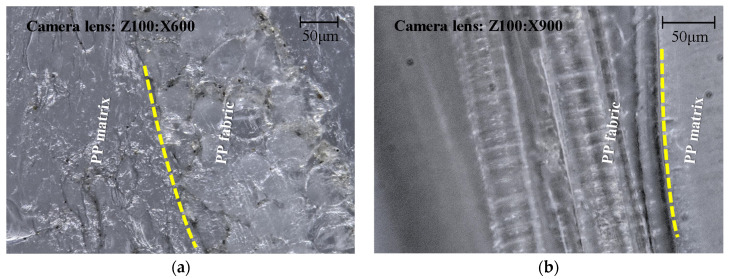
Digital microscopy pictures of the PP SPC sample: (**a**) cross-section of the sample from the side view and (**b**) fracture section of the sample after the tensile test from the top view.

**Table 1 polymers-14-00023-t001:** Material information for the preparation of PP SPC samples.

Component	Material	Brand	Melting Point	Other Properties
Matrix	PP granules	Marlex HGZ-1200	166	Density: 0.907 g/cm^3^, Melt flow rate: 115 g/10 min at 230 °C
Reinforcement	PP fabric	Innegrity LLC	177	Type: plain woven, Areal density: 170 g/m^2^, Diameter of the single fiber: 48 μm

**Table 2 polymers-14-00023-t002:** Injection molding parameters for the preparation of the PP SPC samples.

Experimental No.	Barrel Temperature (°C)	Injection Pressure (MPa)	Injection Speed (m/s)	Holding Time (s)
1	200	127.6	0.13	5
2	220	127.6	0.13	5
3	240	127.6	0.13	5
4	260	127.6	0.13	5
5	280	127.6	0.13	5
6	240	44.8	0.13	10
7	240	58.6	0.13	10
8	240	72.4	0.13	10
9	240	93.1	0.13	10
10	240	113.8	0.13	10
11	240	127.6	0.13	10
12	240	89.6	0.03	10
13	240	89.6	0.08	10
14	240	89.6	0.13	10
15	240	89.6	0.18	10
16	240	89.6	0.23	10
17	240	89.6	0.28	10
18	240	103.4	0.13	1
19	240	103.4	0.13	5
20	240	103.4	0.13	10
21	240	103.4	0.13	15
22	240	103.4	0.13	20
23	240	103.4	0.13	25
24	240	103.4	0.13	60
25	240	103.4	0.13	120
26	260	127.6	0.18	60

**Table 3 polymers-14-00023-t003:** Comparison of optimal tensile strength of PP SPC samples produced by different injection molding methods.

Reference	Method	Type of Reinforcement	Optimal Tensile Strength (MPa)
[6]	Filament winding, compression and pelletizing, injection molding	Chopped pellets	35.6
[8]	Co-extrusion, injection molding	Chopped pellets	30
[9]	Injection compression molding	Plain knitted fabric	32.5
[11]	Insert injection molding	Plain fabric	70
[13]	Insert injection molding	Plain fabric	38
[17]	Melt sequential injection molding	-	55.3
This work	Insert injection molding with optimized processing	Plain fabric	120

## Data Availability

The datasets generated and analyzed during the current study are available from the corresponding author on reasonable request.

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
