# Peer review of "Effects of Injection Molding Parameters on Properties of Insert-Injection Molded Polypropylene Single-Polymer Composites"

_polymers, 2021, doi:10.3390/polym14010023_

Round 1

Reviewer 1 Report

This paper aims to optimize the parameters of the injection molding process of polypropylene mono-polymer composites.
The authors can consider the following aspects:
- The introduction needs to be substantially improved, in the sense that other bibliographic sources in the field need to be considered;
- The properties of the materials in section 2.1 must be presented in tabular form;
- The research methodology and research objective are not clear. The parameter values ​​shown in Table 1 are not justified. A certain method of experiment design must be adopted. Under other conditions, the results obtained may not allow optimization.
- In order to ensure an optimization of the parameters, it is necessary to carry out a statistical processing of the results obtained in the experimental research;
- the first part of section 3.1 must be moved to the materials and methods section;
- macroscopic images of the test specimens obtained must be presented;
- a microscopic analysis of the test material is also required;
- The discussion part is non-existent and the contribution of the research presented in the paper in relation to other research in the field is not highlighted;
- At the end of the conclusions, the future research directions must be specified;
- the conclusions should be more concrete and cover the practical applications of the results obtained

Author Response

This paper aims to optimize the parameters of the injection molding process of polypropylene mono-polymer composites.

The authors can consider the following aspects:

- The introduction needs to be substantially improved, in the sense that other bibliographic sources in the field need to be considered;

Response: We have improved the section of the introduction, referred to more publications, and discussed the research topic mainly related to this work.

- The properties of the materials in section 2.1 must be presented in tabular form;

Response: We have added Table 1 to summarize the material information.

- The research methodology and research objective are not clear. The parameter values ​​shown in Table 1 are not justified. A certain method of experiment design must be adopted. Under other conditions, the results obtained may not allow optimization.

Response: The method used in this work is insert injection molding for the production of SPC parts. As we discussed in the introduction, it is different from traditional compression molding. The advantages of the insert injection molding method are the enlarged processing temperature window and the short cycle time. The objective of this work is to investigate the effects of injection molding parameters on the properties of the SPC parts and find out suitable processing conditions for optimal mechanical strength. We have added some sentences in the introduction.

Therefore, we used the single-factor experimental design as listed in Table 2. We conducted 25 groups of experiments. At least six cycles were done under one group of processing conditions. The best values of the single factors were determined one by one. Finally, the best values of the single factors combined together in Experimental No. 26 to be the optimal conditions to validate the result. The best tensile strength was obtained, which proves the feasibility of the single factor experimental design. We have added some sentences to make it clear.

- In order to ensure optimization of the parameters, it is necessary to carry out a statistical processing of the results obtained in the experimental research;

Response: As we know, the injection molding process is very complex, and it is related to many processing parameters. There may not be only one parameter combination to obtain the optimal strength of PP SPC parts. Furtherly, the effects of these injection molding parameters are usually nonlinear. The method of statistical analysis requires a lot of experiments, and the results might be not accurate.

- the first part of section 3.1 must be moved to the materials and methods section;

Response: We have moved the relative section to 2.2.

- macroscopic images of the test specimens obtained must be presented;

Response: The images of the test specimen, the fabric, and the mold cavity have been given in Figure 2.

- microscopic analysis of the test material is also required;

Response: The microscopic images of the test specimen have been given in Figure 10. The relative discussion has been added.

- The discussion part is non-existent and the contribution of the research presented in the paper in relation to other research in the field is not highlighted;

Response: The section title “Results” should be “Results and discussion”, we have revised this. In order to highlight our research results, we have added the comparison of tensile strength of PP SPCs prepared by different injection molding methods. Table 3 and one paragraph have been added to the revised manuscript.

- At the end of the conclusions, the future research directions must be specified;

Response: We have added the future research work at the end of the conclusions.

- the conclusions should be more concrete and cover the practical applications of the results obtained

Response: The practical applications of the results have been given at the end of the conclusions.

Reviewer 2 Report

The work describes the influence of injection parameters on the strength properties and impact toughness of a material based on PP as a polymer matrix. The filler was PP fiber, which is why we call such a system single-polymer composite.

The work is written correctly, you can pay attention to the presentation of the results.

There is no clear indication of the novelty of this work, the analysis of the results should be more based on a comparison with the results of other scientists - please complete this.

Please expand the introduction to other research papers.

Please improve the English language. 

Author Response

Thank you very much for the review comments. Accordingly, we have revised the manuscript to address the comments from the reviewers and also improved the writing quality. The detailed point-by-point responses are attached as follows: 

The work describes the influence of injection parameters on the strength properties and impact toughness of a material based on PP as a polymer matrix. The filler was PP fiber, which is why we call such a system single-polymer composite.

The work is written correctly, you can pay attention to the presentation of the results.

There is no clear indication of the novelty of this work, the analysis of the results should be more based on a comparison with the results of other scientists - please complete this.

Response: The method used in this work is insert injection molding for the production of SPC parts. As we discussed in the introduction, it is different from traditional compression molding. The advantages of the insert injection molding method are the enlarged processing temperature window and the short cycle time. The objective of this work is to investigate the effects of injection molding parameters on the properties of the SPC parts and find out suitable processing conditions for optimal mechanical strength. We have added some sentences in the introduction.

Please expand the introduction to other research papers.

Response: More related research papers have been added and discussed in the introduction.

Please improve the English language. 

Response: the English language has been improved.

Round 2

Reviewer 1 Report

The authors revised their manuscript according to my suggestions. Thus the manuscript can be accepted for publication.